# 3D-Fast Gray Matter Acquisition with Phase Sensitive Inversion Recovery Magnetic Resonance Imaging at 3 Tesla: Application for detection of spinal cord lesions in patients with multiple sclerosis

**Adrien Goujon** [1]*, **Sonia Mirafzal**[1], **Kevin Zuber**[2], **Romain Deschamps**[3], **Jean-Claude Sadik**[1], **Olivier Gout**[3], **Julien Savatovsky**[1], **Augustin Lecler**[1]

**1** Department of Neuroradiology, Foundation Adolphe de Rothschild Hospital, Paris, France, **2** Department of Clinical Research, Foundation Adolphe de Rothschild Hospital, Paris, France, **3** Department of Neurology, Foundation Adolphe de Rothschild Hospital, Paris, France

* adriengoujon@gmail.com

**Data Availability Statement:** Because of the RGPD policy in France, we are not allowed to upload the

## Abstract

### Background and purpose

To compare 3D-Fast Gray Matter Acquisition with Phase Sensitive Inversion Recovery (3D-FGAPSIR) with conventional 3D-Short-Tau Inversion Recovery (3D-STIR) and sagittal T1- and T2-weighted MRI dataset at 3 Tesla when detecting MS spinal cord lesions.

### Material and methods

This prospective single-center study was approved by an institutional review board and enrolled participants from December 2016 to August 2018. Two neuroradiologists blinded to all data, individually analyzed the 3D-FGAPSIR and the conventional datasets separately and in random order. Discrepancies were resolved by consensus by a third neuroradiologist. The primary judgment criterion was the number of MS spinal cord lesions. Secondary judgment criteria included lesion enhancement, lesion delineation, reader-reported confidence and lesion-to-cord-contrast-ratio. A Wilcoxon's test was used to compare the two datasets.

### Results

51 participants were included. 3D-FGAPSIR detected significantly more lesions than the conventional dataset (344 versus 171 respectively, p<0.001). Two participants had no detected lesion on the conventional dataset, whereas 3D-FGAPSIR detected at least one lesion. 3/51 participants had a single enhancing lesion detected by both datasets. Lesion delineation and reader-reported confidence were significantly higher with 3D-FGAPSIR: 4.5 (IQR 1) versus 2 (IQR 0.5), p<0.0001 and 4.5 (IQR 1) versus 2.5 (IQR 0.5), p<0.0001. Lesion-to-cord-contrast-ratio was significantly higher using 3D-FGAPSIR as opposed to 3D-STIR and T2: 1.4 (IQR 0,3) versus 0.4 (IQR 0,1) and 0.3 (IQR 0,1)(p = 0.04).

data. Additionally, it is considered sensitive data because it cannot be fully anonymized. Data can be requested by contacting Mr. Kevin Joubel from our ethics committee: dpo@for.paris.

**Funding:** Unfunded study.

**Competing interests:** The authors have declared that no competing interests exist.

**Abbreviations:** 3D-FGAPSIR, 3 Dimension-Fast Gray Matter Acquisition with Phase Sensitive Inversion Recovery; LCCR, Lesion to Cord Contrast Ratio; MRI, Magnetic Resonance Imaging; MS, Multiple Sclerosis; STIR, Short-Tau Inversion Recovery; WI, Weighted imaging.

Correlations with clinical data and inter- and intra-observer agreements were higher with 3D-FGAPSIR.

## Conclusion

3D-FGAPSIR improved overall MS spinal cord lesion detection as compared to conventional set and detected all enhancing lesions.

## Introduction

Multiple sclerosis (MS) is an inflammatory disease of the central nervous system affecting both the encephalon and the spinal cord. Spinal cord involvement is detected in up to 68–83% of patients with clinically definite MS on Magnetic Resonance Imaging (MRI), while 7.5–15% of patients with MS have only spinal cord lesions [1]. Spinal cord imaging is recommended by international guidelines [2,3] for diagnosing and managing patients with MS.

However, spinal cord MRI can be challenging due to the spine's small size and image artifacts caused by various impediments, such as proximity to nearby bones or motion from breathing, heartbeats or swallowing. The sensitivity of conventional imaging like T2-weighted imaging (WI) to show spinal cord lesions in MS is low, as demonstrated by radiological-pathological correlations [4]. This low sensitivity on sequences may be related to low lesion to normal cord contrast [5,6]. Furthermore, it might be one of the reasons for the poor correlation between clinical scores and the number of lesions detected.

Optimized MR sequences have been developed and show significant improvement of detection and delineation of lesions, mostly by improving lesion contrast to noise ratio and lesion to cord contrast ratio (LCCR) [5,7–12]. Magnetization prepared rapid gradient echo (MPRAGE) allows for more accurate calculation of lesion load whereas double-inversion-recovery (DIR) provides better spatial delineation of lesions [11,12]. Among them, Phase-Sensitive Inversion-Recovery (PSIR) showed promise by increasing detection of spinal cord lesions and active enhancement in the cord, paving the way for an all-in-one sequence [13,14].

These non-conventional techniques provide major improvements in both the diagnosis and follow-up of patients with MS, since spinal cord is one of the four cardinal lesions in MS. They allow detection of previously non detectable spinal cord lesions.

We optimized the inversion time of a 3D PSIR sequence to null the white matter signal, which aims to improve the lesion to normal cord contrast. This new approach will be referred to in this paper as the 3D-Fast Gray Matter Acquisition with Phase Sensitive Inversion Recovery (FGAPSIR).

The aim of our study was to evaluate the detection of spinal cord lesions by 3D-FGAPSIR as compared to a conventional dataset including post contrast 3D-Short-Tau Inversion Recovery (3D-STIR), sagittal T2- and T1-WI.

## Material and methods

### Ethics committee approval

This prospective single center study was conducted in a tertiary referral center specializing in neurological disease. The ethics committee of Ile-de-France I approved this study and signed informed consent was obtained from all patients. This study follows the strengthening the reporting of observational studies in epidemiology guidelines (STROBE).

## Participants

From December 2016 to August 2018, 57 consecutive MS patients undergoing spine MRI were included. Inclusion criteria were: (a) age over 18 years and (b) confirmed diagnosis of MS as defined by the 2010 McDonald criteria [15]. Participants with any MRI contraindication were not included.

Secondary exclusion criteria were (a) incomplete MR exam defined by the absence of one or several sequences among those tested (b) insufficient quality of MRI for interpretation due to the presence of artifacts. 51 patients were enrolled for analysis. Selection of participants is shown in S1 Fig.

Clinical data (type of MS defined as primary progressive, secondary progressive or relapsing-remitting, disease duration and Expanded Disability Status Scale [EDSS]) were reported by a neurologist specialized in MS with 20 years of experience (BLINDED).

## MRI protocol

All MR Images were acquired with a 3.0 T imager (Philips) with a 16-channel head coil and a posterior spine coil (Philips Medical Systems, Best, The Netherlands).

Two post-contrast imaging datasets were performed. The first one was called a "conventional dataset," including sagittal T2- and T1- weighted imaging (WI) and 3D turbo spin echo-STIR, as recommended by international guidelines [16–19]. The second one was the "3D-FGAPSIR dataset". The 3D-FGAPSIR was developed combining a Fast Grey Matter Acquisition T1 Inversion Recovery and a 3D-PSIR sequence optimized for spinal cord visualization [13,20]. Starting from the 3D-PSIR sequence, we modified the Turbo Field Echo inversion prepulse and the shot duration in order to increase the possible T1 contrast range and to obtain a Fast Grey Matter Acquisition T1 Inversion Recovery contrast on the magnitude image. The shot duration was set at 400ms, as performed for the FGATIR, which improves the lesion to white matter contrast, due to the longer T1 relaxation of MS lesions compared to white matter [20,21]. The advantage of the 3D FGAPSIR is that the range of the T1-weighted is increased, ranging from -1 to 1 instead of 0 to 1 in a T1 SE sequence. This provides improved grey matter-to-white matter contrast with clear lesion delineation as well as hypointense CSF signal intensity because of its large negative magnetization. This offers a good visualization of spinal cord lesions which appear hypointense in the white matter of the spinal cord.

A single acquisition provided two magnitude images, the first one with a Turbo Field Echo inversion prepulse and the second one without it, and one phase-corrected real image calculated using a reference scan, as done in the PSIR sequence. The 3D-FGAPSIR was completed in one acquisition covering the spine from the lower brainstem through the mid-to-lower dorsal spine, whereas the conventional dataset was completed over two acquisitions covering the spine from the lower brainstem through the conus medullaris. Detailed acquisition parameters are displayed in Table 1.

Additional axial T2-WI (TR 4042 ms; TE 120 ms; number of excitations 1; slice thickness 3 mm with no gap; FOV 180 x 180 mm; bandwidth 171 Hz; acquisition matrix 400 x 285; acquisition duration 3 minutes 30 seconds) could be acquired for clinical purposes at the discretion of the radiologist performing the MR acquisitions, but was not included in the reading datasets. All images were acquired 10 minutes after a single bolus (0.1 mmol/kg) of Gadobutrol (Gadovist; Bayer HealthCare, Germany, Berlin). To avoid any effect due to gadolinium impregnation, sequences were acquired in random order.

**Table 1. Detailed MRI acquisition parameters.**

|  | Sagittal T2-WI | Sagittal T1-WI | 3D TSE STIR | 3D-FGAPSIR |
|---|---|---|---|---|
| Scan Mode | 2D | 2D | 3D | 3D |
| Repetition time (ms) | 3000 | 500 | 3500 | 8.6 |
| Echo time (ms) | 100 | 16 | 40 | 4.6 |
| Inversion time (ms) | - | - | 180 | 400 |
| Flip/Refocusing angle | 90˚/120˚ | 80˚/120˚ | 90˚/120˚ | 5˚ (reference image)/8˚/variable |
| Number of excitations | 2 | 2 | 1 | 1 |
| Slice thickness (mm) | 2 | 3 | 1.2 | 1.2 |
| Gap | 0.3 | 0 | - | - |
| Voxel size | 1x1.2 | 1x1.2 | 1.2x1.2x1.2 | 1.2x1.2x1.2 |
| Field of view (mm) | 180x360 | 180x360 | 320x349x69 | 308x207x42 |
| Bandwidth (kHz) | 773 | 260 | 240 | 197.7 |
| Acquisition Matrix | 188x297 | 188x308 | 268x291x57 | 256x172x70 |
| Acquisition duration | 3min45s | 3min20s | 5min18sec | 3min46s |
| TSE/TFE Factor | 29 | 7 | 49 | 58 |
| TFE inversion pre pulse (ms) | - | - | - | 400 |
| TFE shot duration (ms) | - | - | - | 661 |
| TFE shot interval (ms) | - | - | - | 2300 |
| Foldover direction | Feet-Head | Feet-Head | Feet-head | Antero-posterior |
| Oversampling (mm) | 170/170 | 170/170 | 97/154 | - |
| Sense factor | - | 1.3 | 1.7 | 3 |

WI: Weighted Imaging; STIR: Short-Tau Inversion Recovery; FGAPSIR: Fast Gray Matter Acquisition with Phase Sensitive Inversion Recovery; TSE: Turbo Spin Echo; TFE: Turbo Field Echo; -: Not applicable.

## MRI analysis

Two neuroradiologists (BLINDED, BLINDED with 4 and 5 years of experience in neuroradiology, respectively), blinded to clinical data, individually read the randomized results of the conventional dataset and those of the 3D-FGAPSIR. Eight weeks later, a second reading session was performed to analyse intra-observer agreement. Disagreements were resolved by consensus. An additional consensus reading was performed by a neuroradiologist with 9 years of experience (BLINDED), also blinded to clinical data four weeks after the second reading session. During this session, readers looked at the entire imaging dataset with all available sequences, including axial T2-WI if performed, to determine whether the lesions observed in both datasets were "true" lesions. This was considered the reference standard. The results of this consensus session were used to determine sensitivity, specificity, positive predictive value, negative predictive value and accuracy. All reading sessions were made on a dedicated workstation with the HOROS software (Horosproject.org, Nimble Co LLC d/b/a Purview in Annapolis, MD USA).

The readers assessed the following characteristics of participants' MRIs:

- The main criterion was the number of spinal cord lesions, defined as hyperintense lesions on T2-WI and 3D-STIR and hypo or hyper intense lesions on the phase-corrected real image or the second magnitude images of the 3D-FGAPSIR, respectively. Readers were permitted to reformat 3D sequences in all further required planes. 3D-STIR and sagittal-T2-WI were cross-referenced to ease reading and lesion detection detection of lesions.

- The precise level of the lesions in the sagittal plane, according to the related cervical or dorsal vertebral body levels.

- The precise location of the lesions in the axial plane, defined as anterior, posterior, lateral left and lateral right. When a lesion overlapped on more than one location onto one or more locations, only the predominant location was reported.

- The lesion enhancement corresponding to an active lesion, defined as focal hyperintense lesions on the post contrast T1-WI or on 3D-FGAPSIR first magnitude images.

- The lesion delineation measured as follows: 1 corresponded to poor/unreadable delineation, 2 to low, 3 to moderate, 4 to good and 5 to excellent delineation.

- Reader-reported confidence when detecting spinal cord lesions measured as follows: 1 corresponded to very low confidence, 2 to low, 3 to moderate, 4 to high and 5 to very high confidence.

- The presence of artifacts, defined only as elements corrupting the image, excluding all other elements which could prevent visualization of the lesions, such as partial volume averaging or poor contrast. They were assessed according to the following artifact score: 1 corresponded to no artifact, 2 to minor artifacts, 3 to moderate artifacts, 4 to substantial artifacts, and 5 to severe artifacts.

- The lowest spinal cord level covered by these sequences, according based onto the related dorsal vertebral body levels.

## Lesion contrast

Quantitative measurements of MRI signals were obtained by drawing two regions of interest in order to determine quantitative image quality: in the biggest spinal cord lesion (lesion signal) and in the normal-appearing spinal cord (cord signal). The shape, size, and location of the regions of interest were kept constant among all image sets by using coregistration.

Lesion to cord contrast ratio (LCCR) was calculated with the following formula:

$$\text{LCCR} = \frac{|S\ lesion - S\ cord|}{S\ cord}$$

## Statistical analysis

The sample size was calculated based on the minimum expected mean difference of at least one lesion per patient in the 3D-FGAPSIR dataset as compared to the conventional dataset and a common standard deviation of 2.5 lesions. Assumptions were based on data from literature [13,20]. The statistical power was set at 0.8, and the significance criterion was set to 0.05, with a two-tailed analysis. 51 participants would be necessary for this statistical analysis. A final objective of 57 participants was set to anticipate secondary exclusions and unusable data.

Quantitative variables were presented as mean (standard deviation), median (interquartile range or IQR), and categorical variables as percentages. A Wilcoxon's test was used to compare both data sets. Inter and intra-observer agreement for MRI reading was assessed using Interclass Correlation Coefficients (ICC) with a 95% confidence interval and were interpreted as: <0.40 poor, 0.40–0.59 fair, 0.60–0.74 good, 0.75–1 excellent. Spearman's correlation coefficients were calculated to assess the correlation between EDSS Score and the number of lesions with the following criteria being used to interpret the results: <0.30 poor correlation, ≥0.30 and ≤0.70 mild correlation and >0.70 strong correlation [22]. A p-value below 0.05 was considered statistically significant. Data were analyzed using R software [23].

## Results

### Demographics

51 participants were included (34 women and 17 men, mean age 43+/- 22.5 years). Demographic and clinical data are provided in Table 2.

### Lesion detection

3D-FGAPSIR detected significantly more overall lesions than the conventional dataset (344 vs 171, p<0.001) (Figs 1 and 2). 3D-FGAPSIR detected significantly more lesions in both cervical (214 vs 94, p<0.001) and dorsal (130 vs 77, p<0.001) regions (Fig 3). 3D-FGAPSIR detected significantly more anterior (37 versus 5, p<0.001) and lateral (178 versus 47, p<0.001) lesions than the conventional dataset. Posterior lesions were detected similarly in both datasets: 129 versus 119 respectively (S2 Fig).

None of the 3D-FGAPSIR dataset lesions were considered false-positive during the final consensus session. There were 17 missed lesions in 3D-FGAPSIR compared to the conventional dataset, all of them in lower spinal cord levels not covered by the 3D-FGAPSIR sequence. All lesions detected with the conventional dataset above T8 were also visible on 3D-FGAPSIR. Sensitivity, Specificity, Positive Predictive Value, Negative Predictive Value and Accuracy were of 95.3%, 100%, 100%, 95.3% and 97.6% for the 3D-FGAPSIR and 49.7%, 100%, 100%, 66.5% and 74.9% for the conventional dataset respectively.

Two participants (4%) had at least one lesion on 3D-FGAPSIR images not detected on the conventional dataset. Conversely, there were no participants having at least one lesion on the conventional dataset and none on 3D-FGAPSIR.

### Enhancing lesions

3 participants (6%) had a single enhancing lesion. All enhancing lesions were detected on both the 3D-FGAPSIR first magnitude sequence and the sagittal T1-WI from the conventional dataset (Fig 4).

### Lesion delineation and reader-reported confidence

Lesion delineation was significantly higher with 3D-FGAPSIR than with the conventional dataset: median score 4.5 (IQR 1) versus 2 (IQR 0,5) (p<0.0001).

Reader-reported confidence was significantly higher with 3D-FGAPSIR than with the conventional dataset: median confidence 4.5 (IQR 1) versus 2.5 (IQR 0,5) (p<0.0001).

**Table 2. Detailed demographic and clinical data of patients with Multiple Sclerosis (MS).**

|  |  | Relapsing-Remitting MS (n = 38 patients) | Secondary-Progressive MS (n = 8 patients) | Primary-Progressive MS (n = 5 patients) | Overall |
|---|---|---|---|---|---|
| Gender | Men | 13 | 3 | 1 | 17 |
|  | Women | 25 | 5 | 4 | 34 |
| Age (years)(mean [sd]) |  | 42 [21.7] | 47 [26.7] | 37 [18.3] | 43 [22.5] |
| EDSS (median [IQR]) |  | 3 [3.2] | 3 [3.4] | 3 [2.1] | 3 [3.5] |
| Disease Duration (years)(median [IQR]) |  | 6.9 [9.7] | 12.1 [18.7] | 7 [8.7] | 7.5 [10.5] |

sd: Standard deviation; IQR: Interquartile ratio; EDSS: Expanded Disability Status Scale.

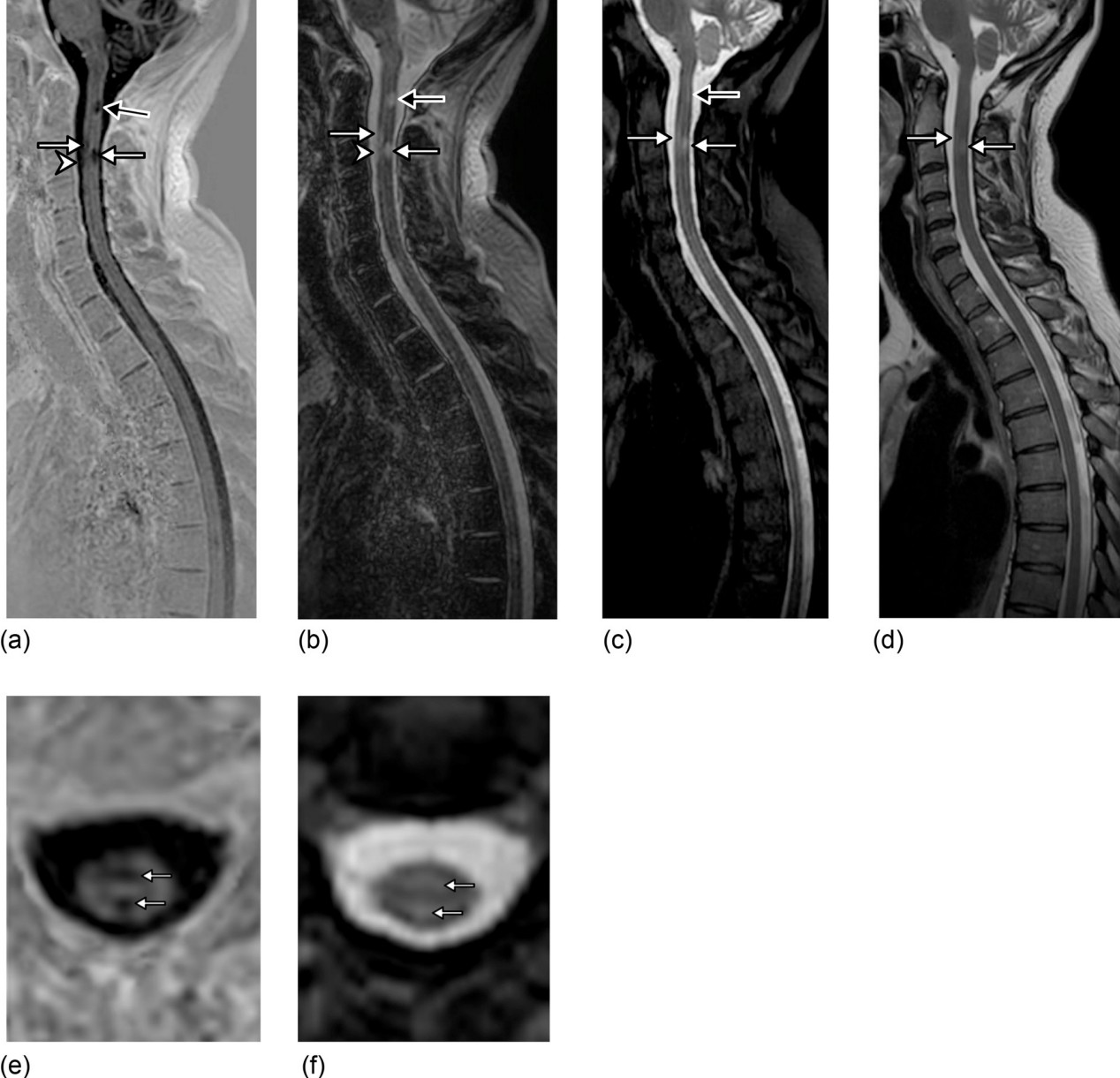

**Fig 1. A 32 year-old woman with relapsing-remitting multiple sclerosis.** 3D-Fast Gray Matter Acquisition with Phase Sensitive Inversion Recovery (3D-FGAPSIR) real-corrected (a) and second magnitude (b) images reformatted in the sagittal plane, providing more accurate detection and delineation of one anterior and one posterior C3 lesions (white arrow) and showing one supplementary distinct anterior C3 lesion (arrowhead) as compared to 3D-Short-Tau Inversion Recovery reformatted in the sagittal plane (c) and sagittal T2-weighted imaging (d). 3D-FGAPSIR also showing a posterior C2 lesion (black arrow) with a high reader-reported confidence, high delineation and high conspicuity, whereas the same lesion is hardly visible on the conventional dataset. 3D-FGAPSIR real-corrected image reformatted in the axial plane (e) localizing and delineating more precisely the C3 lesions as compared to 3D-STIR reformatted in the axial plane (f).

## Presence of artifacts

There was no significant difference for the presence of artifacts between 3D FGAPSIR and the conventional dataset, with a median artifact score of 2. Three patients had severe motion artifacts affecting all the MRI sequences performed indistinctly.

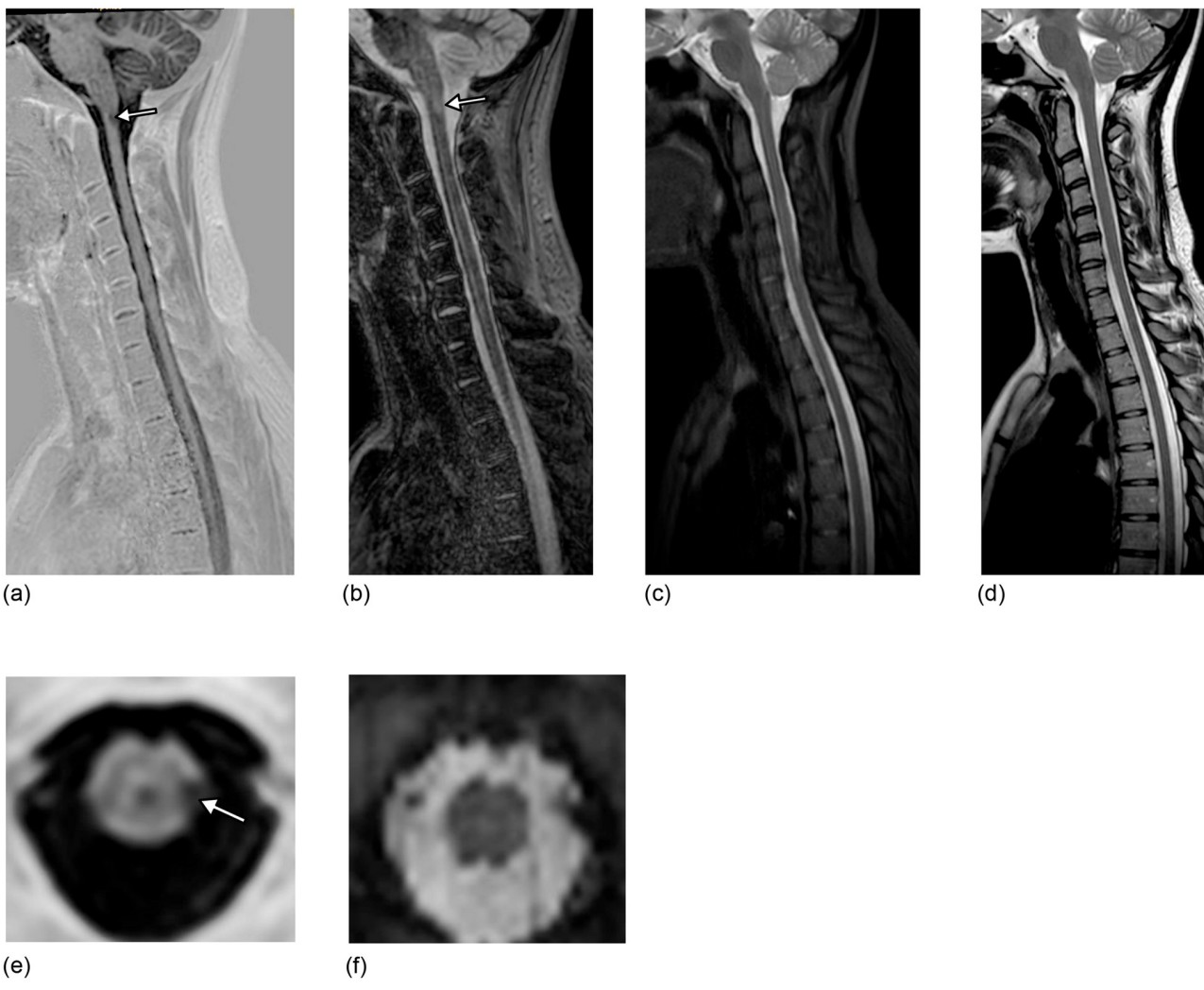

**Fig 2. A 23 year-old man with relapsing-remitting multiple sclerosis.** 3D-Fast Gray Matter Acquisition with Phase Sensitive Inversion Recovery (3D-FGAPSIR) real-corrected (a) and second magnitude (b) images reformatted in the sagittal plane showing a single left posterior cervical spinal cord lesion (arrow), whereas no lesion was detected on both 3D-Short-Tau Inversion Recovery (STIR) reformatted in the sagittal plane (c) or sagittal T2-weighted imaging (d). 3D-FGAPSIR real-corrected image reformatted in the axial plane (e) confirming and precisely localizing the lesion, whereas no lesion was detected on 3D-STIR reformatted in the axial plane (f).

### Lower spinal cord level

The median lower levels covered by the conventional dataset and 3D-FGAPSIR were T12 (+/- 2 levels) and T7 (+/- 1 level) respectively.

### Lesion contrast

LCCR was significantly higher with 3D-FGAPSIR than with 3D-STIR or sagittal T2-WI: 1.4 (IQR 0.3) versus 0.4 (IQR 0.1) and 0.3 (IQR 0.1) (p = 0.004) respectively.

### Inter and intra reader agreement

Overall intra-reader agreement was excellent for both readers with 3D-FGAPSIR: ICC = 0.96 (0.83–0.99) and 0.88 (0.57–0.97), whereas it was only good for the conventional dataset: ICC = 0.64 (0.10–0.92) and 0.75 (0.24–0.94).

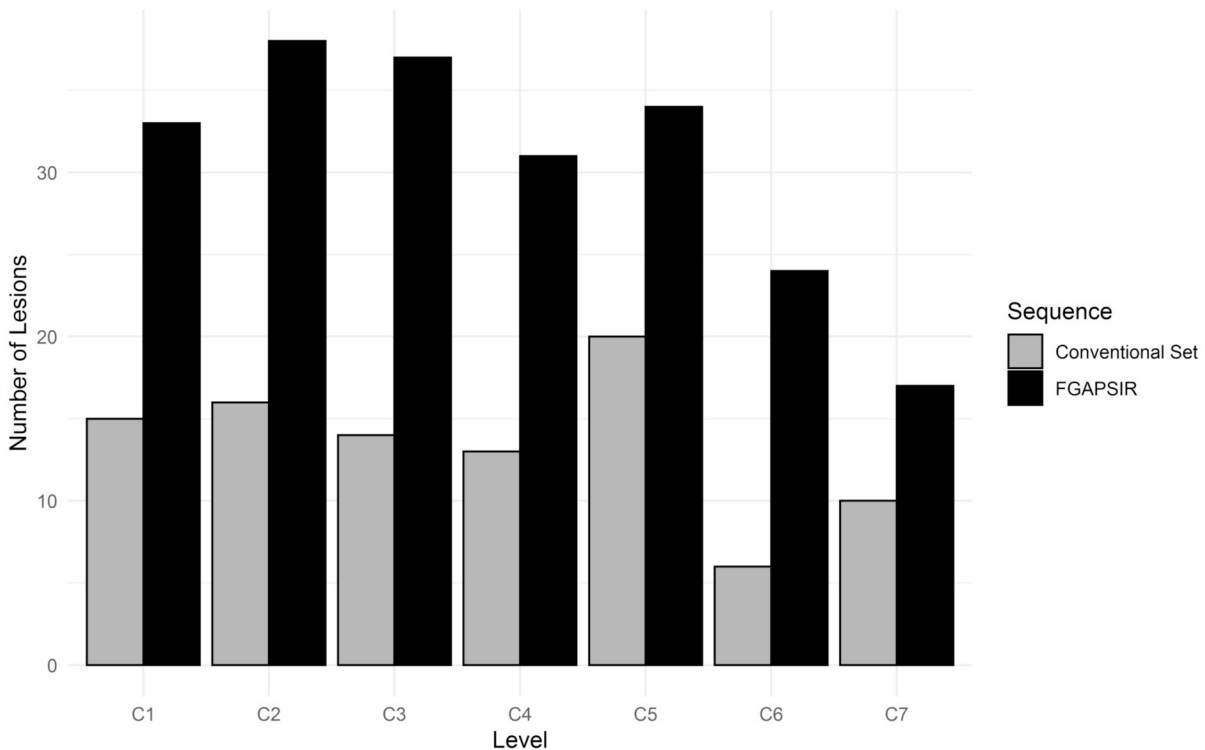

**Fig 3. Comparison of the number and distribution of multiple sclerosis lesions detected in the spinal cord with the conventional imaging dataset (grey) and the 3D-Fast Gray Matter Acquisition with Phase Sensitive Inversion Recovery (3D-FGAPSIR) dataset (black).** The Y-axis represents the number of lesions detected. The X-axis represents the spinal cords levels. The dot below T7 shows the median lower spinal cord level covered by the 3D-FGAPSIR. The horizontal line represents the full range of lower spinal cord levels covered by the 3D-FGAPSIR, with the minimal and maximal lower spinal cord levels represented by the two vertical dotted lines in T5 and T10.

### Correlations with clinical data

There was an overall better correlation between the severity of EDSS scores and the number of lesions detected on 3D-FGAPSIR than on the conventional dataset: 0.38 (p = 0.01) versus 0.29 (p = 0.052), respectively.

There was an overall better correlation between the severity of EDSS scores and the number of lesions detected on 3D-FGAPSIR in progressive stages versus the correlation in RRMS patients: 0.9 versus 0.4.

## Discussion

Our study showed that 3D-FGAPSIR improved overall spinal cord lesion detection in patients with MS with better delineation, higher reader-reported confidence and higher lesion contrast as compared to a conventional dataset including sagittal T2- and T1-WI and 3D turbo spin echo-STIR.

Our significantly higher detection rate is in line with previous studies evaluating new optimized spinal cord sequences showing significantly increased detection rates of MS lesions [5–8,10–14,24]. Similarly to other optimized MR sequences, the 3D-FGAPSIR significantly improves lesion contrast and lesion delineation, reported to be the main determinants increasing detection rates of these sequences. In addition, our study presents several newfound points as compared to similar articles. Our results might be explained by the combination of several improvements of our 3D-FGAPSIR as compared to T2-WI and 3D-STIR. As compared to

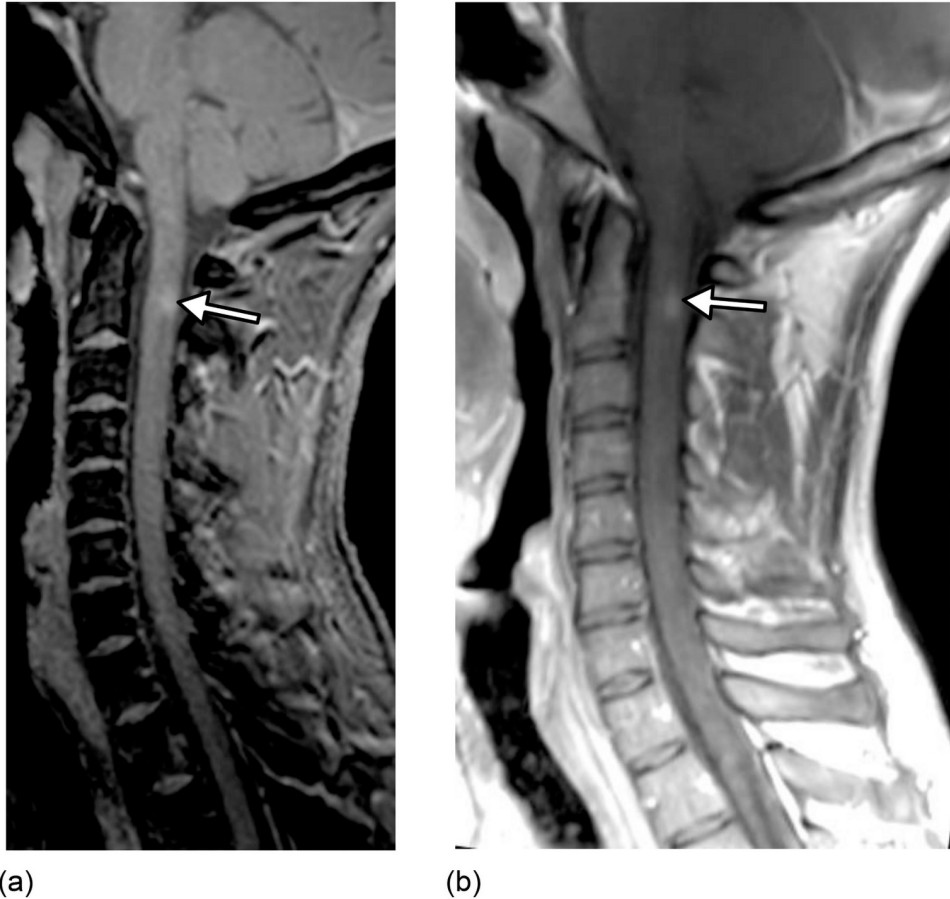

(a) (b)

**Fig 4. A 32 year-old man with relapsing-remitting multiple sclerosis.** 3D-Fast Gray Matter Acquisition with Phase Sensitive Inversion Recovery (3D-FGAPSIR) first magnitude imaging reformatted in the sagittal plane (a) and sagittal T1-weighted imaging (b), showing a single cervical spinal cord enhanced lesion (arrow) corresponding to a clinically active lesion.

both T2 and a 3D STIR similar to the 3D-FGAPSIR for resolution (5,21), 3D-FGAPSIR provided a significantly higher lesion contrast, which is considered a usable reflection of the sequence's ability to detect spinal cord lesions. This was confirmed by significantly higher lesion delineation and reader-reported confidence with 3D-FGAPSIR with the most visible and conspicuous spinal cord lesions. Our results are in line with a recently published study comparing a 3D PSIR sequence with a 3D STIR, showing more than twice as many lesions with the 3D PSIR sequence as opposed to 3D STIR.

3D-FGAPSIR also benefits from the advantages of tri-dimensional acquisition and high resolution, with an isotropic voxel size of 1.2mm$^3$, allowing accurate delineation and localization of spinal cord lesions, without adding any focused axial acquisitions, given that the spinal cord can be analyzed with axial reformats [25]. In our series, 3D-FGAPSIR detected significantly more lesions in anterior and lateral locations, which might reflect the better ability to overcome specific artefacts such as partial volume averaging and cerebrospinal fluid flow, which might mask lesions on conventional sequences.

One other substantial strength of 3D-FGAPSIR might be its ability to identify active enhancing spinal cord lesions. The sequence is primarily T1-weighted which makes it sensitive to gadolinium injection [6,26]. In our study, 3D-FGAPSIR detected all enhancing lesions

visible on T1-WI in three patients. Even if one should remain cautious given the low number of enhancing lesions in our study, it suggests that a unique "all-in-one" 4 minute 3D-FGAPSIR sequence might provide actionable information regarding both spinal cord lesion load and activity simultaneously, thus data regarding dissemination in space and time as well.

Improving detection of spinal cord lesions is clinically very important since the spinal cord is one of the four main locations taken into account for diagnosing MS [3]. Spine involvement and lesion distribution are also highly predictive for conversion to MS in patients with clinically or radiologically isolated symptoms or for clinical disability and are thus major prognostic factors [27–29]. Moreover, quantification of disease activity is important for monitoring treatment efficacy. 3D-FGAPSIR might avoid misclassifying patients as having no evidence of disease activity, which is increasingly considered the treatment goal as with the 2 patients in our series without any lesion detected using the conventional dataset [30–32]. 3D-FGAPSIR might also improve the assessment of disability, as suggested by the better correlation between 3D-FGAPSIR lesion load with EDSS scores as compared to the conventional dataset. This correlation was only fair, which might be due to the fact that EDSS is a composite score reflecting brain and spinal cord involvement and might not be an adequate clinical score when taking into account the spinal cord alone [33]. The correlation was higher in patients with progressive stages versus RRMS patients, suggesting 3D-FGAPSIR might be even more relevant and useful in these patients. Nevertheless, one might expect that 3D-FGAPSIR with its high resolution, increased ability to delineate lesions and high contrast, could be used for accurate evaluation of lesion burden, allowing for more accurate correlation with symptoms and disability. It might also help to compare longitudinal MRIs and to count accurately spinal cord lesions during follow-up because a more precise coregistration can be performed with 3D sequences as compared to those with 2D.

The duration of our 3D-FGAPSIR was 3 minutes 46 seconds, which is compatible with clinical practice and well-adapted to patients with MS with spinal cord lesions who may be less likely to hold still for long periods of time. It was substantially faster than the conventional imaging dataset which had an overall duration time of more than 12 minutes.

Our study suffers from several limitations. Firstly, our patient population was relatively small and imaging was performed in a single center. The assumptions on overall spinal cord detection on which we calculated the sample size proved to be correct, but we lacked data to analyze precisely the performance of 3D-FGAPSIR, as in the case of detecting enhancing spinal cord lesions. However, our study was a prospective and controlled study. This study design decreases potential bias and strengthens our results. Therefore, it provides higher quality evidence regarding the superiority of the 3D-FGAPSIR sequence.

Secondly, 3D-FGAPSIR did not cover the whole spinal cord, missing coverage under the T8 level for the majority of our patients, as compared to the conventional dataset covering both lower spinal cord and conus medullaris. This lack of coverage could have provided an underestimation of lesions detected. This issue could be overcome by adding a second acquisition up to the conus medullaris.

Thirdly, readers noted a substantial increase of spinal cord artifacts within the lowest spinal cord levels covered by the 3D-FGAPSIR, especially in larger patients. These artifacts might be resolved by performing 3D-FGAPSIR acquisitions with a craniocaudal phase, by oversampling and by enlarging the antero-posterior field of view in order to avoid wrap-around artifacts at the abdominal levels. However, these artifacts remained minor. 3D-FGAPSIR showed significantly higher lesion delineation and reader-reported confidence scores, lesion contrast and inter and intra reader agreement, as compared to the conventional dataset.

Fourthly, there is no reference standard for identifying "true" spinal cord lesions and it was impossible to have a real reference standard since this would imply post-mortem histological

exams. However, white matter lesions in other neurological diseases are reported to be uncommon in the spinal cord [34] and strong histopathologic data support the accuracy of previously optimized spinal cord sequences [35]. Nevertheless, we cannot be sure that all lesions detected were, in fact, MS lesions. The sensitivity, specificity, positive predictive value, negative predictive value and accuracy that we reported were based on a careful consensus reading, but might be inaccurate.

Fifthly, analytical optimization could be performed to improve the lesion to white matter contrast of the FGAPSIR sequence. Using our TR and TE parameters, the inversion time might be optimal between 500-600ms to allow an efficient suppression of the white matter, and thus an increase of the lesion to white matter contrast.

Sixthly, we did not compare the FGAPSIR sequence with the original PSIR sequence to demonstrate the FGAPSIR improves lesion detection.

Finally, readers knew which method they were assessing because each sequence had easily-recognizable features, which could have led to a certain bias.

## Conclusion

Our study showed that 3D-FGAPSIR improved overall spinal cord lesion detection in patients with MS with better delineation, higher reader-reported confidence and higher lesion contrast as compared to a conventional dataset including sagittal T2- and T1- WI and 3D turbo spin echo-STIR.

## Supporting information

**S1 Fig. Flow chart.** MS: Multiple Sclerosis; MRI: Magnetic Resonance Imaging.
(PDF)

**S2 Fig. Bar charts showing the detailed locations of the lesions seen by the 3D-FGAPSIR sequence and missed by the conventional dataset in the sagittal (a) and axial planes (b).** y-axis indicate the number of lesions.
(PDF)

**S1 Table. Comparison between secondary excluded patients and patients enrolled in the final study cohort.** sd: standard deviation; IQR: Interquartile ratio; EDSS: Expanded Disability Status Scale.
(DOCX)

**S2 Table. Detailed locations of spinal cord lesions according to the type of Multiple Sclerosis (MS).**
(DOCX)

## Acknowledgments

Laura McMaster provided professional English-language medical editing of this article.

## Author Contributions

**Conceptualization:** Julien Savatovsky, Augustin Lecler.

**Data curation:** Adrien Goujon, Sonia Mirafzal.

**Formal analysis:** Adrien Goujon, Kevin Zuber.

**Investigation:** Olivier Gout.

**Methodology:** Kevin Zuber, Romain Deschamps, Olivier Gout, Julien Savatovsky.

**Resources:** Romain Deschamps.

**Supervision:** Jean-Claude Sadik.

**Writing – original draft:** Adrien Goujon, Augustin Lecler.

**Writing – review & editing:** Julien Savatovsky, Augustin Lecler.

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
