## [Decision Letter · Decision Letter 0]

23 Sep 2020

PONE-D-20-19595

3D-Fast Gray Matter Acquisition with Phase Sensitive Inversion Recovery Magnetic Resonance Imaging at 3 Tesla: an innovative MRI sequence used to detect spinal cord lesions in patients with multiple sclerosis

PLOS ONE

Dear Dr. Goujon,

Thank you for submitting your manuscript to PLOS ONE. After careful consideration, we feel that it has merit but does not fully meet PLOS ONE’s publication criteria as it currently stands. Therefore, we invite you to submit a revised version of the manuscript that addresses the points raised during the review process.

I would like to thank you for your patience during the review process. The two reviewers ended up with conflicting recommendations and rather than delay the decision further by recruiting a third reviewer, I have opted for a major revision. I would like to encourage you to pay particular attention to address the methodological and validation concerns raised by Reviewer 1.

We look forward to receiving your revised manuscript.

Kind regards,

Niels Bergsland

Academic Editor

PLOS ONE

Journal Requirements:

3. Please respond by return e-mail with an updated version of your manuscript to amend either the abstract on the online submission form or the abstract in the manuscript so that they are identical. We can make any changes on your behalf.

4. Please include your tables as part of your main manuscript and remove the individual files. Please note that supplementary tables (should remain/ be uploaded) as separate "supporting information" files

Reviewers' comments:

Reviewer's Responses to Questions

**Comments to the Author**

1. Is the manuscript technically sound, and do the data support the conclusions?

Reviewer #1: No

Reviewer #2: Yes

2. Has the statistical analysis been performed appropriately and rigorously? 

Reviewer #1: Yes

Reviewer #2: Yes

3. Have the authors made all data underlying the findings in their manuscript fully available?

Reviewer #1: Yes

Reviewer #2: Yes

4. Is the manuscript presented in an intelligible fashion and written in standard English?

Reviewer #1: Yes

Reviewer #2: Yes

5. Review Comments to the Author

Reviewer #1: In this work the authors propose the use of a new fast 3D PSIR sequence for the detection of multiple sclerosis (MS) lesions in the spinal cord. The application of such a sequence in the brain and also in the spinal cord has been already shown to be effective for the detection of MS lesions. In particular, in the spinal cord, this sequence has been suggested also for the segmentation of the grey matter tissue. Authors starts from these same evidences to motivate the application of this sequence in MS with some innovations. However, although the title suggests some technical innovations, these are not well introduced and explained. The details of the changes are referred to other two papers, but it remains unclear what is the original idea of the present work. If the novelty is the adjustment of the inversion time to null the signal from the white matter, as done in reference 13 for an MPRAGE sequence, this should be better clarified and the reason why this should improve contrast between lesions and the surrounding tissue proven. In fact, it is not obvious that zeroing the signal from the white matter will increase the contrast with lesions. Especially for a paper suggesting technical advances, the inversion time should be optimized to this aim analytically, for instance, and the new solution validated by comparing the contrast with the original PSIR, to demonstrate that this solution is effective. It should also be considered that when the TI is set to nulled the white matter all the other relevant tissues, having a longer T1, are acquired when the magnetization is still along the negative y axis and the expanded contrast of a phase contrast image preserved; so I am wondering about the need of obtaining a reference scan, as done in this PSIR sequence, for correcting the background phase variations.

Another point is about the comparison with the 3D STIR sequence, similar to the proposed sequence for resolution, but also in the way the contrast is constructed: the discrepancy between the number of lesions identified is really high.

All these aspects should be better considered in the discussion of the paper from a technical point of view. Overall I found the discussion to optimistic about the potentiality of the sequence and I found some conclusions, such that of suggesting its use for the atrophy measurements or that that this single sequence could replace the whole recommended protocol for the spinal cord in MS, too simplistic.

These are other additional comments:

1. In the introduction it is stated that “the sensitivity of conventional imaging like T2-weighted imaging to show spinal cord lesions is low” This sentence simplifies the concept without giving details. Could authors elaborate a little more on this?

2. “Optimized MR sequences have been developed and show significant improvement of detection and

delineation of lesions” Following point 1, could authors explain the major improvements introduced in these non- conventional techniques?

3. One of the exclusion criteria was the insufficient quality of MRI for the interpretation, due to the presence of artifacts. However the robustness against artifacts should be an important characteristic to be reported and analyses. Authors should describe these cases instead.

4. MRI protocol. It seems that all sequences were acquired after contrast injection. The international guidelines prescribe a pre-contrast protocol followed by a post contrast T1 weighted sequence, in case there are T2-hyper-intense visible lesions. Please explain your choice of acquiring all sequences after contrast injection. Furthermore I would add the direct reference for the recommended MR protocol in MS, since the cited paper refers to other papers about this point.

5. Sequence parameters: for the PSIR sequence the flip angle for the reference image is usually shorter than that of the inversion recovery part of the sequence. If this is the case, the second flip angle should be specified. For the 3D STIR sequence, was a variable flip angle used to compensate for the long echo train required to keep acquisition time short enough? Could this have affected the quality of the images, because of the induced filtering effect?

6. In the discussion it is stated that characteristic of the sequence is the increased contrast of spinal cord lesions by combining the benefits of nulling normal white matter, suppression of surrounding CSF signals. This is not totally true or at least this is not necessarily the optimal condition for the contrast. Also, in this sequence, CSF is not nulled, but the signal appears dark because of the negative values.

Reviewer #2: Summary

The authors present prospective single-center data that convincingly demonstrate spinal cord lesion detection via 3D-Fast Gray Matter Acquisition with Phase Sensitive Inversion Recovery (3D-FGAPSIR) is improved as compared to conventional 3D-Short-Tau Inversion Recovery (3D-STIR) and sagittal T1- and T2-weighted sequences at 3 Tesla in 51 subjects with MS. 3D-FGAPSIR detected more lesions than the conventional methods and offered improved lesion delineation and lesion-to-cord contrast ratio. In general, the study is well designed, the observations are novel and clinically relevant. I think the study could be published with some minor revisions.

Comments by Section

1. Methods, Participants, Page 6: Did the 6 omitted subjects differ from the larger group in any meaningful way (age, sex, disease category, MRI features)? This could be added to the Supplementary Data.

2. Methods, MRI Analysis, Page 8-9: Have similar lesion delineation and reader-reported confidence scales been used in prior work? Would cite if so.

3. Results, Lesion Detection Page 11: Did lesion detection via 3D-FGAPSIR outperform conventional imaging differentially based on MS disease stage. For example, with longstanding disease cord lesions made become less apparent over time. Some kind of analysis of this breakdown might be a useful addition. As a suggestion, the lesion data (cervical/dorsal, anterior/lateral) presented in the results here might also be shown in a separate table similar to table 2, comparing RRMS, SPMS, PPMS and Overall.

4. Results, Lesion Detection Page 11: It would be interesting to report the details of the lesions seen by 3D-FGAPSIR that conventional imaging missed. Consider showing this in a Figure.

5. Results, “Correlation with clinical data”: Analysis of relationship to clinical status could be strengthened by other outcomes that may better reflect damage to the cord. Analysis of association to ambulatory walk time if performed, EDSS subscores (sensorimotor function, bowel bladder function vs. cerebral, cerebellar, brainstem), for example might be means of expanding that clinical analysis. I appreciate that this data may not be available and such analysis beyond the scope of this imaging study.

6. Results, “Correlation with clinical data”: Similar to prior comment, EDSS linkage to lesions could be assessed for RRMS vs. progressive stages perhaps to shed light on 3D-FGAPSIR clinical utility.

7. Discussion, Line 5: The Discussion could expand upon how the improvement in lesion detection by FGAPSIR over conventional methods compares to other similar technical advances noted int the studies cited here.

8. Tables & Figures: No major concerns. My only comment would be that panels/images of the axial reconstructed view may helpful to demonstrate the author’s assertion regarding the superiority of FGAPSIR over conventional techniques for anterior and lateral locations.

Minor Comments/Grammar Suggestions

1. Introduction, Page 5, Line 6: Minor grammatical edit. Would suggest use of the phrase “can be challenging” rather than “remains a challenge” and delete “to” in the phrase “to image artifacts”.

2. Introduction, Page 5 Line 10: I would clarify “spinal cord” lesion burden and “in MS”.

3. Introduction, Page 5 Line 13-14: Delete the word “showing” here, change phrasing to “active enhancement in the cord”.

4. Introduction, Page 5 Line 16: Use “performance” (singular). This is duplicated in the Discussion, Page 13, Line 15.

5. Introduction, Page 5 Line 19: The aim was to “evaluate the detection of spinal cord lesions by 3D-FGAPSIR as compared to…”

6. Methods, Page 7, MRI Analysis. Line 3: “Height” (spelling).

7. Methods, Page 7, MRI Analysis. Line 6: “neuroradiologist” (singular).

8. Methods, Page 7, MRI Analysis. Line 7: Delete the word “later”.

9. Methods, Page 8, MRI Analysis. Line 4: positive and negative predictive value need not be capitalized. This is repeated in the final paragraph of the Discussion as well.

10. Methods, Page 8, MRI Analysis. Line 17-18: Correct the spacing “locationonto” and would delete word “most” from “most predominant”.

11. Results, Lesion Detection: The subtitle should be capitalized consistent, so “Lesion detection".

12. Discussion, Page 14, Line 19: Change phrasing to “3D-FGAPSIR might also improve the assessment of disability”.

13. Discussion, Page 15, Line 6: patients who “may be less likely to hold still”.

6. PLOS authors have the option to publish the peer review history of their article (what does this mean?). If published, this will include your full peer review and any attached files.

Reviewer #1: No

Reviewer #2: No

---

## [Author Response · Author response to Decision Letter 0]

17 Nov 2020

Reviewer #1:

R1.1 : In this work the authors propose the use of a new fast 3D PSIR sequence for the detection of multiple sclerosis (MS) lesions in the spinal cord. The application of such a sequence in the brain and also in the spinal cord has been already shown to be effective for the detection of MS lesions. In particular, in the spinal cord, this sequence has been suggested also for the segmentation of the grey matter tissue. Authors starts from these same evidences to motivate the application of this sequence in MS with some innovations. However, although the title suggests some technical innovations, these are not well introduced and explained. The details of the changes are referred to other two papers, but it remains unclear what is the original idea of the present work. If the novelty is the adjustment of the inversion time to null the signal from the white matter, as done in reference 13 for an MPRAGE sequence, this should be better clarified and the reason why this should improve contrast between lesions and the surrounding tissue proven. 

We thank the Reviewer for his/her comment. We agree that the technical innovations associated with the FGAPSIR sequence were not sufficiently well explained. Both the introduction and methods have been rephrased to clarify the modifications performed on the FGAPSIR sequence, as follows:

 Introduction

“Our center developed a 3T MRI high resolution sequence which aims to improve the lesion to normal cord contrast.”

 Methods

“The 3D-FGAPSIR was developed combining a Fast Grey Matter Acquisition T1 Inversion Recovery (FGATIR, 13) and a 3D-PSIR sequence optimized for spinal cord visualization (11). Starting from the 3D-PSIR sequence, we modified the Turbo Field Echo inversion prepulse and the shot duration in order to increase the possible T1 contrast range and to obtain a Fast Grey Matter Acquisition T1 Inversion Recovery contrast on the magnitude image. The shot duration was set at 400ms, as performed for the FGATIR (13), which improves the lesion to white matter contrast, due to the longer T1 relaxation of MS lesions compared to white matter (Lommers et al,2019).”

Lommers, E., Simon, J., Reuter, G., Delrue, G., Dive, D., Degueldre, C., ... & Maquet, P. (2019). Multiparameter MRI quantification of microstructural tissue alterations in multiple sclerosis. NeuroImage: Clinical, 23, 101879. doi: 10.1016/j.nicl.2019.101879

In fact, it is not obvious that zeroing the signal from the white matter will increase the contrast with lesions. Especially for a paper suggesting technical advances, the inversion time should be optimized to this aim analytically, for instance, and the new solution validated by comparing the contrast with the original PSIR, to demonstrate that this solution is effective.

The shot duration was set at 400ms following the parameters of the cited papers (Fechner et al, 2019; Sudhyadhom et al, 2009), which have already demonstrated an improvement of the lesion to white matter contrast, as well as increased detection of MS lesions (Fechner et al, 2019). However, we agree with the Reviewer that analytical optimization could be performed to improve the lesion to white matter contrast. This optimization could be performed following the signal equation of an IR sequence and an inversion time chosen to null the signal intensity amplitude of the white matter:

S_IR (T1,TE,TR)=ρ(1-2e^(-TI/T1)+2e^(-((TR-TE/2))/T1)-e^(-TR/T1))e^(-TE/T2)

Hou, P., Hasan, K. M., Sitton, C. W., Wolinsky, J. S., & Narayana, P. A. (2005). Phase-sensitive T1 inversion recovery imaging: a time-efficient interleaved technique for improved tissue contrast in neuroimaging. American journal of Neuroradiology, 26(6), 1432-1438.

Using our TR and TE, the inversion time might be optimal between 500-600ms to allow an efficient suppression of the white matter (figure below), and thus an increase of the lesion to white matter contrast. This optimization needs more investigation, and the sequences parameters might be optimized in the following studies. 

We agree that a direct comparison with the original PSIR is relevant. However, our study was a preliminary study whose aim was to validate the FGAPSIR as compared to a conventional imaging dataset. Further studies will be conducted to compare the FGAPSIR and the original PSIR sequences. 

We highlighted these points in the Limitations section of our manuscript.

It should also be considered that when the TI is set to nulled the white matter all the other relevant tissues, having a longer T1, are acquired when the magnetization is still along the negative y axis and the expanded contrast of a phase contrast image preserved; so I am wondering about the need of obtaining a reference scan, as done in this PSIR sequence, for correcting the background phase variations.

Indeed, the FGAPSIR phase-corrected image was calculated using a reference scan, as done in the PSIR sequence. We highlighted this point in the Material and Methods section.

R1.2 Another point is about the comparison with the 3D STIR sequence, similar to the proposed sequence for resolution, but also in the way the contrast is constructed: the discrepancy between the number of lesions identified is really high.

All these aspects should be better considered in the discussion of the paper from a technical point of view. 

We agree that the discrepancy between the number of lesions identified with the 3D STIR and the 3D FGAPSIR is high. As you mentioned, the 3D STIR sequence we performed in our study matched the 3D FGAPSIR for resolution and met the technical requirements recommended by the international guidelines. Moreover, our results are in line with a recently published study comparing a 3D PSIR sequence with a 3D STIR (Mirafzal et al 2020), showing more than twice as many lesions with the 3D PSIR sequence. We highlighted this point in the discussion section.

Mirafzal S, Goujon A, Deschamps R, Zuber K, Sadik JC, Gout O, Lecler A, Savatovsky J. 3D PSIR MRI at 3 Tesla improves detection of spinal cord lesions in multiple sclerosis. J Neurol. 2020 Feb;267(2):406-414. doi: 10.1007/s00415-019-09591-8. Epub 2019 Oct 26.)

R1.3. Overall I found the discussion too optimistic about the potentiality of the sequence and I found some conclusions, such that of suggesting its use for the atrophy measurements or that this single sequence could replace the whole recommended protocol for the spinal cord in MS, too simplistic.

We agree that our study was a preliminary study and that the discussion might be too optimistic. As suggested, we deleted the sentences not directly supported by the results of our study.

These are other additional comments:

R1.4 In the introduction it is stated that “the sensitivity of conventional imaging like T2-weighted imaging to show spinal cord lesions is low” This sentence simplifies the concept without giving details. Could authors elaborate a little more on this?

As requested, we elaborated more on this statement in the introduction section.

R1.5 “Optimized MR sequences have been developed and show significant improvement of detection and delineation of lesions” Following point 1, could authors explain the major improvements introduced in these non- conventional techniques?

We added a sentence in the introduction section to explain the major improvements introduced in these non- conventional techniques.

R 1.6 One of the exclusion criteria was the insufficient quality of MRI for the interpretation, due to the presence of artifacts. However the robustness against artifacts should be an important characteristic to be reported and analyses. Authors should describe these cases instead.

We agree the robustness against artifacts is an important characteristic. Therefore, we provided data regarding the presence of artifacts in the Results section. We added a sentence in the Material and Methods section accordingly.

R1.7 MRI protocol. It seems that all sequences were acquired after contrast injection. The international guidelines prescribe a pre-contrast protocol followed by a post contrast T1 weighted sequence, in case there are T2-hyper-intense visible lesions. Please explain your choice of acquiring all sequences after contrast injection. Furthermore I would add the direct reference for the recommended MR protocol in MS, since the cited paper refers to other papers about this point.

All sequences were acquired after contrast injection, which is the protocol for all our patients with MS. The occurrence of a pre contrast T1 hyper intense lesion is exceedingly rare, and our protocol is designed to increase the delay between gadolinium injection and completion of the MRI sequences, which allows for better detection of MS lesions. This protocol follows international guidelines. 

As requested, we added the direct references for recommended MR protocols in MS. 

R1.8. Sequence parameters: for the PSIR sequence the flip angle for the reference image is usually shorter than that of the inversion recovery part of the sequence. If this is the case, the second flip angle should be specified. For the 3D STIR sequence, was a variable flip angle used to compensate for the long echo train required to keep acquisition time short enough? Could this have affected the quality of the images, because of the induced filtering effect?

As requested we specified the second flip angle in Table 1. The flip angle for the reference image is shorter than that of the inversion recovery part of the sequence.

Regarding the 3D STIR sequence, the flip angle was kept constant to obtain the best quality image possible.

R1.9. In the discussion it is stated that characteristic of the sequence is the increased contrast of spinal cord lesions by combining the benefits of nulling normal white matter, suppression of surrounding CSF signals. This is not totally true or at least this is not necessarily the optimal condition for the contrast. Also, in this sequence, CSF is not nulled, but the signal appears dark because of the negative values.

We agree this statement was imprecise. We modified the discussion accordingly. 

Reviewer #2: Summary

The authors present prospective single-center data that convincingly demonstrate spinal cord lesion detection via 3D-Fast Gray Matter Acquisition with Phase Sensitive Inversion Recovery (3D-FGAPSIR) is improved as compared to conventional 3D-Short-Tau Inversion Recovery (3D-STIR) and sagittal T1- and T2-weighted sequences at 3 Tesla in 51 subjects with MS. 3D-FGAPSIR detected more lesions than the conventional methods and offered improved lesion delineation and lesion-to-cord contrast ratio. In general, the study is well designed, the observations are novel and clinically relevant. I think the study could be published with some minor revisions.

Comments by Section

R2.1 Methods, Participants, Page 6: Did the 6 omitted subjects differ from the larger group in any meaningful way (age, sex, disease category, MRI features)? This could be added to the Supplementary Data.

There were no significant differences between the 6 omitted subjects and the final study cohort. We provided Supplementary Table 1 to report these results.

R2.2 Methods, MRI Analysis, Page 8-9: Have similar lesion delineation and reader-reported confidence scales been used in prior work? Would cite if so.

To the best of our knowledge, similar lesion delineation and reader-reported confidence scales have not been used in prior work.

R2.3 Results, Lesion Detection Page 11: Did lesion detection via 3D-FGAPSIR outperform conventional imaging differentially based on MS disease stage. For example, with longstanding disease cord lesions made become less apparent over time. Some kind of analysis of this breakdown might be a useful addition. As a suggestion, the lesion data (cervical/dorsal, anterior/lateral) presented in the results here might also be shown in a separate table similar to table 2, comparing RRMS, SPMS, PPMS and Overall.

We agree that this analysis is valuable. As we enrolled a majority of patients with RRMS and only 13 patients with progressive stages in this prospective study, our statistician was unable to conduct statistic studies on such small samples. Therefore, we provided only raw data regarding the number of spinal cord lesions detected by each dataset according to the type of MS in Supplementary Table 2.

R2.4. Results, Lesion Detection Page 11: It would be interesting to report the details of the lesions seen by 3D-FGAPSIR that conventional imaging missed. Consider showing this in a Figure.

As requested, we provided Supplementary Figure 2 to report the details of these missed lesions.

R2.5. Results, “Correlation with clinical data”: Analysis of relationship to clinical status could be strengthened by other outcomes that may better reflect damage to the cord. Analysis of association to ambulatory walk time if performed, EDSS subscores (sensorimotor function, bowel bladder function vs. cerebral, cerebellar, brainstem), for example might be means of expanding that clinical analysis. I appreciate that this data may not be available and such analysis beyond the scope of this imaging study.

We agree that such analysis might be valuable. Unfortunately, we collected only EDSS when enrolling patients in our prospective study, thus this data is not available. However, we intend to launch a more comprehensive validation study including detailed clinical scores and scales to look for clinico-radiological correlations. 

R2.6. Results, “Correlation with clinical data”: Similar to prior comment, EDSS linkage to lesions could be assessed for RRMS vs. progressive stages perhaps to shed light on 3D-FGAPSIR clinical utility.

There was an overall better correlation between the severity of EDSS scores and the number of lesions detected on 3D-FGAPSIR in progressive stages versus in RRMS patients: 0.9 versus 0.4. However, as mentioned before, we enrolled a majority of patients with RRMS and only 13 patients with progressive stages in this prospective study, limiting our ability to conduct statistical studies on such small samples. As recommended by our statistician, we indicated raw correlation values only without associating a statistical test to them. We modified the Results section and the discussion section.

R2.7. Discussion, Line 5: The Discussion could expand upon how the improvement in lesion detection by FGAPSIR over conventional methods compares to other similar technical advances noted in the studies cited here.

As requested, we modified the discussion to elaborate on that point.

R2.8. Tables & Figures: No major concerns. My only comment would be that panels/images of the axial reconstructed view may helpful to demonstrate the author’s assertion regarding the superiority of FGAPSIR over conventional techniques for anterior and lateral locations.

We added axial reconstructed views in the Figure 1, in addition to the ones displayed in Figure 2, to highlight the superiority of FGAPSIR over conventional techniques. 

Minor Comments/Grammar Suggestions

R2.9. Introduction, Page 5, Line 6: Minor grammatical edit. Would suggest use of the phrase “can be challenging” rather than “remains a challenge” and delete “to” in the phrase “to image artifacts”.

We modified the sentence.

R2.10. Introduction, Page 5 Line 10: I would clarify “spinal cord” lesion burden and “in MS”.

As requested, we clarified the sentence.

R2.11. Introduction, Page 5 Line 13-14: Delete the word “showing” here, change phrasing to “active enhancement in the cord”.

As requested, we modified the sentence.

R2.12. Introduction, Page 5 Line 16: Use “performance” (singular). This is duplicated in the Discussion, Page 13, Line 15.

We corrected this error.

R2.13. Introduction, Page 5 Line 19: The aim was to “evaluate the detection of spinal cord lesions by 3D-FGAPSIR as compared to…”

As requested we modified the sentence.

R2.14. Methods, Page 7, MRI Analysis. Line 3: “Height” (spelling).

We corrected this typo.

R2.15. Methods, Page 7, MRI Analysis. Line 6: “neuroradiologist” (singular).

We corrected this typo.

R2.16. Methods, Page 7, MRI Analysis. Line 7: Delete the word “later”.

As requested we deleted the word later.

R2.17. Methods, Page 8, MRI Analysis. Line 4: positive and negative predictive value need not be capitalized. This is repeated in the final paragraph of the Discussion as well.

We modified the sentences in both methods and the discussion sections.

R2.18. Methods, Page 8, MRI Analysis. Line 17-18: Correct the spacing “locationonto” and would delete word “most” from “most predominant”.

We corrected the sentence.

R2.19. Results, Lesion Detection: The subtitle should be capitalized consistent, so “Lesion detection".

We corrected this inconsistency.

R2.20. Discussion, Page 14, Line 19: Change phrasing to “3D-FGAPSIR might also improve the assessment of disability”.

As requested, we modified the sentence.

R2.21. Discussion, Page 15, Line 6: patients who “may be less likely to hold still”.

As requested, we modified the sentence.

---

## [Decision Letter · Decision Letter 1]

3 Dec 2020

PONE-D-20-19595R1

3D-Fast Gray Matter Acquisition with Phase Sensitive Inversion Recovery Magnetic Resonance Imaging at 3 Tesla: an innovative MRI sequence used to detect spinal cord lesions in patients with multiple sclerosis

PLOS ONE

Dear Dr. Goujon,

Thank you for submitting your manuscript to PLOS ONE. After careful consideration, we feel that it has merit but does not fully meet PLOS ONE’s publication criteria as it currently stands. Therefore, we invite you to submit a revised version of the manuscript that addresses the points raised during the review process.

Although several of the issues were sufficiently addressed, Reviewer 1 believes that additional modifications are necessary. Upon re-reading the revised manuscript, I share both of these concerns as well and think that the manuscript can be improved if these points are addressed.

We look forward to receiving your revised manuscript.

Kind regards,

Niels Bergsland

Academic Editor

PLOS ONE

Reviewers' comments:

Reviewer's Responses to Questions

**Comments to the Author**

1. If the authors have adequately addressed your comments raised in a previous round of review and you feel that this manuscript is now acceptable for publication, you may indicate that here to bypass the “Comments to the Author” section, enter your conflict of interest statement in the “Confidential to Editor” section, and submit your "Accept" recommendation.

Reviewer #1: (No Response)

Reviewer #2: All comments have been addressed

2. Is the manuscript technically sound, and do the data support the conclusions?

Reviewer #1: Partly

Reviewer #2: Yes

3. Has the statistical analysis been performed appropriately and rigorously? 

Reviewer #1: (No Response)

Reviewer #2: Yes

4. Have the authors made all data underlying the findings in their manuscript fully available?

Reviewer #1: Yes

Reviewer #2: Yes

5. Is the manuscript presented in an intelligible fashion and written in standard English?

Reviewer #1: Yes

Reviewer #2: Yes

6. Review Comments to the Author

Reviewer #1: Although authors tried to address my concerns, the answers provided and the changes made to the manuscript do not sufficiently clarify the points of my questions. It is still not evident the key idea for the technical improvement presented, that the contrast is optimally maximized with this solution and the need of applying the phase sensitization in this particular case, when the magnetization of the relevant tissues is still along the negative axis. The new reference included (Mirafzal et al. J Neurol. 2020) has already shown the advantage of using the 3DPSIR sequence against a 3D STIR sequence; so what is missing is the advantage of using a PSIR sequence with the proposed inversion time, optimized to suppress the WM, versus the conventional 3D PSIR.

Overall I am satisfied with the answers to my questions, except for questions 1 and 5, that should be reconsidered.

Reviewer #2: The authors have submitted a revision that addresses reviewer suggestions and strengthens the manuscript. suggestions from the initial critique could not be addressed due to statistical power or study design, and the authors discuss these limitations. The manuscript should be published. The following additions offer improvement on the initial submission:

- Newly added Supplementary Table 1 compares features of included/excluded subjects.

- Supplementary Table 2 updated to show disease stage.

- Supplementary Figure 2 shows examples case of lesions seen by 3D-FGAPSIR that conventional imaging missed.

- The Discussion was modified in response to the point related to disease stage correlation between the severity of EDSS scores and the number of lesions detected on 3D-FGAPSIR. The authors found an overall better correlation between the severity of EDSS scores and the number of lesions detected on 3D-FGAPSIR in progressive stages versus in RRMS patients: 0.9 versus 0.4 though opt to present only raw data given that the subgroup analysis is underpowered (13 patients with progressive MS)

(R2.6. Results, “Correlation with clinical data”: Similar to prior comment, EDSS linkage to lesions could be assessed for RRMS vs. progressive stages perhaps to shed light on 3D-FGAPSIR clinical utility.)

- The Discussion was expanded in a thoughtful way that addresses the critique, noting how lesion detection by FGAPSIR offers improvement to conventional and similar alternative methods.

- Additional axial reconstructed images strengthen the case examples illustrated in Figure 1.

- All minor grammatical issues have been corrected.

7. PLOS authors have the option to publish the peer review history of their article (what does this mean?). If published, this will include your full peer review and any attached files.

Reviewer #1: No

Reviewer #2: No

---

## [Author Response · Author response to Decision Letter 1]

9 Jan 2021

Review Comments to the Author :

Reviewer #1: Although authors tried to address my concerns, the answers provided and the changes made to the manuscript do not sufficiently clarify the points of my questions. It is still not evident the key idea for the technical improvement presented, that the contrast is optimally maximized with this solution and the need of applying the phase sensitization in this particular case, when the magnetization of the relevant tissues is still along the negative axis. The new reference included (Mirafzal et al. J Neurol. 2020) has already shown the advantage of using the 3DPSIR sequence against a 3D STIR sequence; so what is missing is the advantage of using a PSIR sequence with the proposed inversion time, optimized to suppress the WM, versus the conventional 3D PSIR.

Overall I am satisfied with the answers to my questions, except for questions 1 and 5, that should be reconsidered.

Reviewer #2: The authors have submitted a revision that addresses reviewer suggestions and strengthens the manuscript. suggestions from the initial critique could not be addressed due to statistical power or study design, and the authors discuss these limitations. The manuscript should be published. The following additions offer improvement on the initial submission:

- Newly added Supplementary Table 1 compares features of included/excluded subjects.

- Supplementary Table 2 updated to show disease stage.

- Supplementary Figure 2 shows examples case of lesions seen by 3D-FGAPSIR that conventional imaging missed.

- The Discussion was modified in response to the point related to disease stage correlation between the severity of EDSS scores and the number of lesions detected on 3D-FGAPSIR. The authors found an overall better correlation between the severity of EDSS scores and the number of lesions detected on 3D-FGAPSIR in progressive stages versus in RRMS patients: 0.9 versus 0.4 though opt to present only raw data given that the subgroup analysis is underpowered (13 patients with progressive MS)

(R2.6. Results, “Correlation with clinical data”: Similar to prior comment, EDSS linkage to lesions could be assessed for RRMS vs. progressive stages perhaps to shed light on 3D-FGAPSIR clinical utility.)

- The Discussion was expanded in a thoughtful way that addresses the critique, noting how lesion detection by FGAPSIR offers improvement to conventional and similar alternative methods.

- Additional axial reconstructed images strengthen the case examples illustrated in Figure 1.

- All minor grammatical issues have been corrected.

 

Reviewer #1: Questions 1 & 5:

Question 1: In this work the authors propose the use of a new fast 3D PSIR sequence for the detection of multiple sclerosis (MS) lesions in the spinal cord. The application of such a sequence in the brain and also in the spinal cord has been already shown to be effective for the detection of MS lesions. In particular, in the spinal cord, this sequence has been suggested also for the segmentation of the grey matter tissue. Authors starts from these same evidences to motivate the application of this sequence in MS with some innovations. However, although the title suggests some technical innovations, these are not well introduced and explained. The details of the changes are referred to other two papers, but it remains unclear what is the original idea of the present work. If the novelty is the adjustment of the inversion time to null the signal from the white matter, as done in reference 13 for an MPRAGE sequence, this should be better clarified and the reason why this should improve contrast between lesions and the surrounding tissue proven. 

R1.1 The advantage of the 3D FGAPSIR is that the range of the T1-weighted is increased, ranging from -1 to 1 instead of 0 to 1 in a T1 SE sequence. This provides improved grey matter-to-white matter contrast with clear lesion delineation as well as hypointense CSF signal intensity because of its large negative magnetization. This offers a good visualization of spinal cord lesions which appear hypointense in the white matter of the spinal cord. We developed the Material and Methods section to explain more in details the technical advantages of the 3D FGAPSIR sequence.

In the proposed sequence, the signal was not optimally maximized as the TI was not perfectly set to suppress the WM. This part needs prospective study to properly set the inversion. Moreover, we totally agree with the Reviewer, that a study investigating the advantage of using the proposed PSIR sequence versus the conventional 3D PSIR is needed. We mentioned these limitations in the limitation section.

Question 2: “Optimized MR sequences have been developed and show significant improvement of detection and delineation of lesions” Following point 1, could authors explain the major improvements introduced in these non- conventional techniques?

R1.5 We modified and expanded the introduction section to explain more in details the major improvements introduced in these non- conventional techniques, from both technical and clinical points of view.

---

## [Decision Letter · Decision Letter 2]

9 Feb 2021

PONE-D-20-19595R2

3D-Fast Gray Matter Acquisition with Phase Sensitive Inversion Recovery Magnetic Resonance Imaging at 3 Tesla: an innovative MRI sequence used to detect spinal cord lesions in patients with multiple sclerosis

PLOS ONE

Dear Dr. Goujon,

Thank you for submitting your manuscript to PLOS ONE. After careful consideration, we feel that it has merit but does not fully meet PLOS ONE’s publication criteria as it currently stands. Therefore, we invite you to submit a revised version of the manuscript that addresses the points raised during the review process.

Specifically, the Reviewer feels that some of the responses to the previous review are inadequate. Although the Reviewer believes that the paper can be published, please try to tighten up your responses/revisions as best as possible as suggested by the Reviewer.

We look forward to receiving your revised manuscript.

Kind regards,

Niels Bergsland

Academic Editor

PLOS ONE

Reviewers' comments:

Reviewer's Responses to Questions

**Comments to the Author**

1. If the authors have adequately addressed your comments raised in a previous round of review and you feel that this manuscript is now acceptable for publication, you may indicate that here to bypass the “Comments to the Author” section, enter your conflict of interest statement in the “Confidential to Editor” section, and submit your "Accept" recommendation.

Reviewer #1: (No Response)

2. Is the manuscript technically sound, and do the data support the conclusions?

Reviewer #1: Partly

3. Has the statistical analysis been performed appropriately and rigorously? 

Reviewer #1: Yes

4. Have the authors made all data underlying the findings in their manuscript fully available?

Reviewer #1: Yes

5. Is the manuscript presented in an intelligible fashion and written in standard English?

Reviewer #1: Yes

6. Review Comments to the Author

Reviewer #1: I appreciate the changes made and the attempt of better explaining the technical aspects. However these are still insufficient in my opinion for a technical paper. My suggestion is to remove the sentences that emphasize technical improvements and innovations.

For instance the title could be changed in this way:

"3D-Fast Gray Matter Acquisition with Phase Sensitive Inversion Recovery Magnetic Resonance Imaging at 3 Tesla: application for detection of spinal cord lesions in patients with multiple sclerosis"

At the end of the introduction I would change the sentence "Our center developed a 3T MRI high resolution sequence" in something like "We optimised the inversion time of an MPRAGE sequence to null the white matter ...^

I would recommend not to mention that the contrast between lesions and the background was optimised, because this is not. This is a finding, not an aim of the study. I am referring to the sentence in the discussion when Authors write "3D-FGAPSIR was specifically targeted to increase the contrast of spinal cord lesions by combining ..."

In the same way the other similar sentence along the manuscript.

7. PLOS authors have the option to publish the peer review history of their article (what does this mean?). If published, this will include your full peer review and any attached files.

Reviewer #1: No

---

## [Author Response · Author response to Decision Letter 2]

15 Feb 2021

Review Comments to the Author :

Reviewer #1: I appreciate the changes made and the attempt of better explaining the technical aspects. However these are still insufficient in my opinion for a technical paper. My suggestion is to remove the sentences that emphasize technical improvements and innovations.

1.1 For instance the title could be changed in this way:

"3D-Fast Gray Matter Acquisition with Phase Sensitive Inversion Recovery Magnetic Resonance Imaging at 3 Tesla: application for detection of spinal cord lesions in patients with multiple sclerosis"

R1.1 The title was changed as requested.

1.2 At the end of the introduction I would change the sentence "Our center developed a 3T MRI high resolution sequence" in something like "We optimised the inversion time of an MPRAGE sequence to null the white matter …

R1.2: As suggested, we have modified this sentence to “We optimized the inversion time of a 3D PSIR sequence to null the white matter signal, which aims to improve the lesion to normal cord contrast.”

1.3 I would recommend not to mention that the contrast between lesions and the background was optimised, because this is not. This is a finding, not an aim of the study. I am referring to the sentence in the discussion when Authors write "3D-FGAPSIR was specifically targeted to increase the contrast of spinal cord lesions by combining ..."

In the same way the other similar sentence along the manuscript.

R1.3: As suggested, we deleted the sentence mentioning the optimization of the contrast between lesions and the background in the discussion.

---

## [Editor Report · Decision Letter 3]

16 Feb 2021

3D-Fast Gray Matter Acquisition with Phase Sensitive Inversion Recovery Magnetic Resonance Imaging at 3 Tesla: application for detection of spinal cord lesions in patients with multiple sclerosis

PONE-D-20-19595R3

Dear Dr. Goujon,

We’re pleased to inform you that your manuscript has been judged scientifically suitable for publication and will be formally accepted for publication once it meets all outstanding technical requirements.

Kind regards,

Niels Bergsland

Academic Editor

PLOS ONE
---

## [Editor Report · Acceptance letter]

8 Apr 2021

PONE-D-20-19595R3 

3D-Fast Gray Matter Acquisition with Phase Sensitive Inversion Recovery Magnetic Resonance Imaging at 3 Tesla: application for detection of spinal cord lesions in patients with multiple sclerosis

Dear Dr. Goujon:

I'm pleased to inform you that your manuscript has been deemed suitable for publication in PLOS ONE. Congratulations! Your manuscript is now with our production department. 

Kind regards, 

on behalf of

Dr. Niels Bergsland 

Academic Editor

PLOS ONE